# Dialect Transfer for Swiss German Speech Translation

**Claudio Paonessa[1], Yanick Schraner[1], Jan Deriu[2], Manuela Hürlimann[2].**
**Manfred Vogel[1], Mark Cieliebak[2]**
[1]University of Applied Sciences and Arts Northwestern Switzerland, Windisch
[2]Zurich University of Applied Sciences, Winterthur
claudio.paonessa@fhnw.ch, deri@zhaw.ch, yanick.schraner@fhnw.ch, hueu@zhaw.ch,
manfred.vogel@fhnw.ch, ciel@zhaw.ch

## Abstract

This paper investigates the challenges in building Swiss German speech translation systems, specifically focusing on the impact of dialect diversity and differences between Swiss German and Standard German. Swiss German is a spoken language with no formal writing system, it comprises many diverse dialects and is a low-resource language with only around 5 million speakers. The study is guided by two key research questions: how does the inclusion and exclusion of dialects during the training of speech translation models for Swiss German impact the performance on specific dialects, and how do the differences between Swiss German and Standard German impact the performance of the systems? We show that dialect diversity and linguistic differences pose significant challenges to Swiss German speech translation, which is in line with linguistic hypotheses derived from empirical investigations.

## 1 Introduction

There are three main challenges when building Swiss German speech-to-text systems. First, Swiss German is a spoken language with no formal writing system. Thus, the task is formulated as the translation of Swiss German audio into Standard German text (Plüss et al., 2021) without access to an intermediate textual representation of the source language. This leads to difficulties where Swiss German and Standard German differ, e.g. in the usage of tenses (where Swiss German does not use preterite) or lexical items that are distinct between the two languages. The second challenge is that Swiss German consists of many dialects that tend to differ. This issue makes training a speech translation system from Swiss German to Standard German a task of translating many dialects into a single language. The third challenge is that Swiss German is a low-resource language with only around 5 Million speakers. Furthermore, the fact that some

dialects differ significantly yields even fewer speakers for each dialect (e.g., Valais only has about 80K speakers).

These three challenges motivate the investigation of the following two questions:

- How does the inclusion and exclusion of dialects during the training of speech translation (ST) models for Swiss German impact the performance on specific dialects?
- Do the differences between Swiss German and Standard German negatively impact the performance of the ST systems?

The first question investigates the issue of the diversity of dialects in a low-resource setting, while the second question investigates the differences between Swiss German and Standard German.

**Contributions.** To answer these questions, we first review the Swiss German dialect landscape and the differences to Standard German. In particular, we devise a set of hypotheses from the literature stating which dialects are expected to differ from each other and which Swiss German phenomena are expected to impact the performance. We then investigate empirically whether the hypotheses match the results of training ST models in various settings.

Our findings show that the empirical results follow the linguistic investigations. That is, there are dialects that we expect to differ significantly from others, which is confirmed by our empirical results. Furthermore, the differences between Swiss German and Standard German impact the performance, where the past tense has the highest impact.

## 2 Swiss German Dialects

### 2.1 Linguistic background

"Swiss German" commonly refers to the dialects spoken in German-speaking Switzerland. All

Swiss German dialects are Alemannic dialects[1] belonging to the High German languages within the West Germanic language family. The Alemannic dialects in Switzerland can be further split into High Alemannic and Highest Alemannic variants[2].

The sociolinguistic situation in German-speaking Switzerland is particular: unlike in other linguistic areas, high prestige is associated with dialects (Ender and Kaiser, 2009), which are considered important markers of regional identity. This means that Swiss German dialects are spoken in most everyday situations, including formal ones. They are, however, only written in informal contexts, such as social media/text messages, and sometimes also for advertisement purposes. On the other hand, formal correspondence, laws, and newspapers are always written in Standard German. The Standard German of Switzerland differs from the varieties in Germany and Austria and is therefore often referred to as "Swiss Standard German."

## 2.2 Differences between Swiss German and Standard German

Alemannic dialects (AL), including Swiss German, differ from Standard German (DE) in several ways such as:

**Phonology:** Middle German monophthongs and diphthongs are preserved in Swiss German (AL "huus" vs DE "Haus" (house) and AL "[liəb]" vs DE "[liːb]" (dear)) and most dialects have completed the High German consonant shift (AL "[ˈxaʃtə]" vs DE "[ˈkastn]" (box)). Stress is placed on the first syllable of a word in Swiss German more often than in DE.

**Grammar:** AL nouns have no genitive case but rather use possessive constructions ("s huus vom buur" - "the house of the farmer" and "im buur sis huus" - "the farmer his house") and the accusative and nominative cases are conflated (except for personal pronouns). The verbal system has no preterite tense, perfect constructs are used instead. Relative clauses always use the particle "wo", and some verbs reduplicate in present infinite form when forming a complex predicate with another verb (e.g. "du lohsch mi lo ässe" - "you let me [let] eat").

**Lexis:** There are a rather large number of Swiss

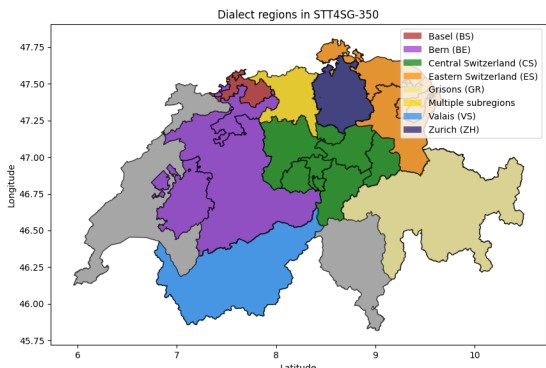

Figure 1: Map showing the seven dialect regions in STT4SG-350.

German / Alemannic vocabulary items which are not intelligible to speakers of Standard German[3]. In Swiss German, some of these originate from French (e.g., "trottoir" for pavement). Further, a highly typical Swiss German word formation process is "-li" suffixation (e.g., "Hündli" - little dog).

In this work, we will investigate the impact of two differences between Swiss German and Standard German: first, preterite tense, which does not exist in Swiss German (Section 6.1), and second, Swiss German vocabulary items that are expected to notably differ from Standard German (Section 6.2).

## 2.3 Dialect regions and dialect differences

The STT4SG-350 speech translation corpus (Plüss et al., 2023a), which we use for the experiments reported in this paper, defines seven dialect regions. They correspond to Swiss cantons as follows:

- **Basel (BS)**: Basel-City, Basel-Country and parts of Aargau
- **Berne (BE)**: Berne, Fribourg, Solothurn and parts of Aargau
- **Central Switzerland (CS)**: Lucerne, most of Schwyz, Ob- and Nidwalden, Uri, Glarus and parts of Aargau
- **Eastern Switzerland (ES)**: St.Gallen, Thurgau, Schaffhausen, Appenzell Innerrhoden and Appenzell Ausserrhoden
- **Grisons (GR)**: Grisons
- **Valais (VS)**: Wallis
- **Zurich (ZH)**: Zurich, Höfe district of Schwyz and parts of Aargau

Figure 1 visualizes these seven dialect regions. We will investigate if similarities between the di-

---

[1]Except for the dialect of Samnaun in Grisons, which is a variant of the Bavarian language.

[2]The traditional dialect of Basel City has many features of Low Alemannic German, but due to internal migration, these have mostly leveled and are now close to High Alemannic.

[3]See, e.g., https://de.wikipedia.org/wiki/Schweizerdeutsch#Wortschatz

alect regions impact ST performance. To quantify the differences between the dialects, we use dialectometric data from the Dialect Atlas of German-Speaking Switzerland (DAGSS; (Hotzenköcherle et al., 1962-1997)) presented by (Scherrer and Stoeckle, 2016). The digitised DAGSS data set, version 3 (DDAGSS), consists of 289 linguistic features (107 phonological, 118 morphosyntactic, and 64 lexical features) collected from local respondents in a field survey across 565 Swiss locations.

We use the DDAGSS features to calculate linguistic distance indices based on the Relative Identity Value (RIV) metric (Goebl, 2010). To apply RIV to the STT4SG-350 data, we match the DDAGGS survey sites to our seven dialect regions, then calculate RIV for all site pairs[4] and average this for all region pairs. The result when using all DDAGSS features[5] is shown in Figure 2. From the pairwise distances, we derive the following hypotheses:

- **VS-hyp.** VS is the most distant dialect from other dialect regions, which corresponds well with local perceptions. Thus, we expect that systems that are trained only on VS perform badly on other dialects, and systems not trained on VS will not perform well on VS.
- **CS/ZH-central-hyp.** CS and ZH have less pronounced differences from most other dialects. Thus, we expect systems trained only on ZH or CS to work better than systems trained only on other dialects.
- **ZH/CS-ES-BS-hyp.** ZH has a very small distance from BS, CS, and ES, respectively. Thus, we expect that systems trained on only BS, CS, or ES, respectively, will perform well on ZH.
- **BE/ES-GR-hyp.** BE has a larger distance to ES and GR than to the other dialects. Thus, we expect systems trained on BE to perform poorly on ES and GR and vice versa.

## 3 Related Work

**General Dialect Transfer.** Luo et al. (2021) and Hassan et al. (2022) showed that fine-tuning a pre-trained English Automatic Speech Recognition

---

[4]Specifically, we use the following process to calculate RIV: for two survey sites, we first identify all features for which they both have a response (*total*). We then count all features where the responses are not identical (*different*). We finally calculate the distance metric as RIV = *different* / *total*

[5]In Appendix C, we display the distance matrices per feature group.

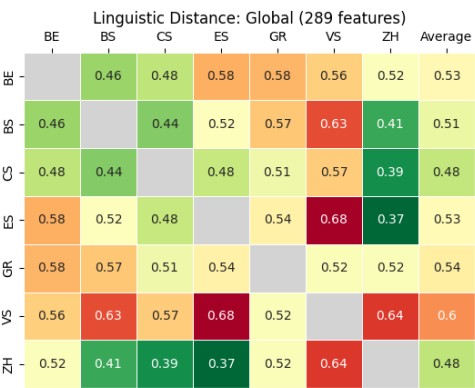

Figure 2: Matrix of linguistic distances between dialect regions.

(ASR) model to different English accents leads to faster training and higher accuracy than a model trained from scratch. Transfer learning from English ASR to different languages such as German, Spanish, and Russian also leads to a higher accuracy than training from scratch (Luo et al., 2021). In (Woldemariam, 2020), transfer learning for low-resource speech recognition in the case of Amharic from English and Mandarin has been studied. For both pre-training languages fine-tuning ASR for Amharic led to an absolute WER reduction of 14.22% and 10.25%, respectively. In our work, we extend the research on transfer learning for different dialects from the ASR set to a speech translation task.

**Swiss German Speech Translation.** Recent high quality datasets such as SwissDial (Dogan-Schönberger et al., 2021), SPC (Plüss et al., 2021), and SDS-200 (Plüss et al., 2022) enabled notable advancements in Swiss German Speech Translation (ST). These datasets successfully cover a wide range of Swiss German dialects. However, we are not aware of any comprehensive investigation into the interactions between these dialects in ASR or ST.

## 4 General Setup

### 4.1 Data

For this investigation, we rely on the STT4SG-350 corpus (Plüss et al., 2023a), which consists of 343 hours of speech translation data. That is, it contains audio in Swiss German and the corresponding Standard German sentence. The sentences were presented to speakers who were asked to translate them to their dialect (without writing down the

| | train_all (bal) | valid | test | full |
|---|---|---|---|---|
| **Hours** | 276 (239) | 34 | 34 | 343 |
| **Rec.** | 200K (173K) | 23K | 25K | 248K |
| **Unique sent.** | 192K (167K) | 23K | 4K | 218K |
| **Speakers** | 219 (192) | 21 | 76 | 316 |
| **Avg. Rec./speaker** | 912 (902) | 1106 | 324 | 783 |

Table 1: Corpus statistics per split. For the train set, the balanced (bal) version is in parentheses. Taken from (Plüss et al., 2023a).

translation) and record them. One effect of this setting is that speakers may use synonyms and different tenses, word order, etc. when saying the sentence aloud in their dialect, leading to a deviation from the reference sentence. The corpus consists of a train and test set. The particular setting of the test set is that it contains 5 hours of audio for each of the seven dialect regions (cp. Section 2.3) based on the *same* set of sentences. This allows for conclusive comparisons of dialect transfer effects. STT4SG-350 also provides a balanced training split with 34 hours for each dialect region (see Table 1). In the train set, the sentences are diverse across the various regions. Furthermore, STT4SG-350 is also balanced with respect to gender and age, achieving an almost uniform distribution across these dimensions.

### 4.2 Models

For the subsequent experiments, we used three models to ensure that the results generalize to various architectures.

**XLS-R.** The first model is a pre-trained XLS-R-300M model (Babu et al., 2021). We initially conducted two different XLS-R-300M experiments, with CTC decoding and with a sequence-to-sequence architecture. The experiments reported here used the CTC architecture, as they perform better, and the sequence-to-sequence results do not provide any additional insights. A comparison of CTC and sequence-to-sequence results is reported in Appendix B.

**Trafo.** The second model is a randomly initialized transformer model implemented in the FAIRSEQ S2T library (Ott et al., 2019) and replicating the model of (Plüss et al., 2022), which we trained from scratch on our data.

**Whisper.** The third model is Whisper small (Radford et al., 2022), a pre-trained transformer-based sequence-to-sequence model.

For the investigation, we relied on medium-sized models: the XLS-R model has 317M parameters, the Trafo model 72M parameters, and Whisper small 244M parameters. We avoid larger models as they make the experiments prohibitively time and cost intensive[6].

## 5 Transfer Experiments

We are interested in the interplay of the dialects: which dialects benefit from other dialects being in the training data, and are there dialects where specialized fine-tuning is more adequate? For this, we run two experiments:

1) Leave-One-Out (LOO), where we train the models on all dialects except one and measure the impact on the performance of the left-out dialect as well as on the other dialects,

2) Single-Dialect (SD), where we fine-tune the models on a single dialect and measure the impact on the performance of the various dialects.

For both experiments, we compare the results to the *All-dialects* setting, which consists in training the model on the full STT4SG-350 corpus, i.e., on all dialects. For all experiments, we report the BLEU score (Papineni et al., 2002), which is calculated using the script provided by (Plüss et al., 2023b). To analyse the difference between *All-dialects* and the different settings, we compute the ratio between the BLEU score achieved by the *All-dialects* setting and the BLEU score achieved by the specific experiment (retainment ratio). We visualise this performance retainment using heatmaps, which show the percentage of performance retained on the $j^{th}$ (column) when leaving out the $i^{th}$ (row) dialect. Thus, a value of 0.9 is to be read as the model achieving 90% of the BLEU score that *All-dialects* achieved. The average at the end of each row indicates the average retainment of the other dialects (i.e., excluding the value of the diagonal), which summarizes the influence of one dialect on other dialects. The absolute values are provided in Table 2.

### 5.1 Leave-One-Out

The leave-one-out (LOO) experiment fine-tunes the models with the data of all dialects while leaving one dialect out. This tests the impact one dialect has on the others through ablation. Table 2 presents the BLEU scores of the LOO experiment. For each dialect column, the LOO row shows the performance of the LOO model on this dialect's test set

---

[6]See Appendix A for a comparison of the 300M and the 1B model of XLS-R, which show a robust correlation between the two models on dialect-specific metrics, indicating that the relative outcomes are independent of model size.

| Model | Experiment | Full | BS | BE | GR | CS | ES | VS | ZH |
|---|---|---|---|---|---|---|---|---|---|
| XLS-R | All dialects | 70.9 | 69.8 | 68.4 | 72.7 | 72.9 | 69.5 | 70.6 | 72.5 |
|  | Leave One Out | 70.4 | 67.7 | 64.9 | 71.0 | 72.3 | 67.4 | 61.2 | 72.2 |
|  | Single Dialect | 57.6 | 68.7 | 65.5 | 69.3 | 71.2 | 69.0 | 71.8 | 69.5 |
| Trafo | All dialects | 62.0 | 59.1 | 57.5 | 64.8 | 65.9 | 62.1 | 61.1 | 63.8 |
|  | Leave One Out | 57.8 | 50.5 | 49.9 | 59.3 | 62.2 | 56.2 | 47.7 | 59.0 |
|  | Single Dialect | 0.1 | 0.1 | 0.1 | 0.1 | 0.2 | 0.1 | 0 | 0 |
| Whisper | All dialects | 62.9 | 60.0 | 59.7 | 64.6 | 65.9 | 62.8 | 62.2 | 65.2 |
|  | Leave One Out | 60.2 | 55.4 | 54.6 | 64.6 | 66.3 | 61.2 | 51.9 | 63.5 |
|  | Single Dialect | 45.4 | 49.5 | 50.5 | 53.2 | 57.4 | 50.1 | 54.9 | 52.9 |

Table 2: BLEU scores of all experiments. Each experiment is a tuple of model and experiment type. There are three types: All-dialects, Leave-One-Out, and Single Dialect. For each setting, we report the score achieved on each dialect separately, and the average score achieved on the full test set.

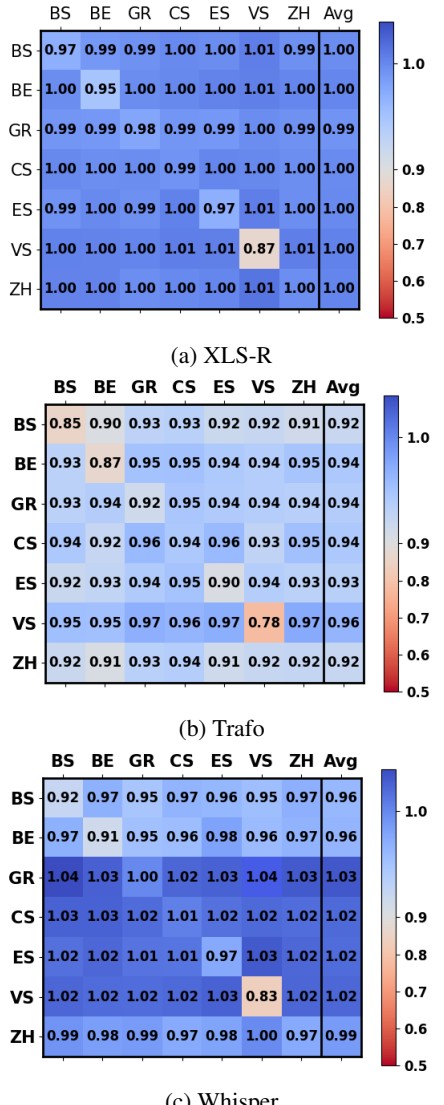

(a) XLS-R

(b) Trafo

(c) Whisper

Figure 3: Results of LOO Experiment: heatmap with retainment ratio. Each row shows the scores achieved for each dialect when leaving out the dialect of the row. The last column shows the average score of the ratios (the average excludes the diagonal values.)

(so when BS is left out, the BLEU score on the BS test set is 67.7). It is apparent (and expected) that

leaving out a dialect leads to lower performance on the left-out dialect compared to using the full dataset. For a deeper analysis of the dependence of the dialects, Figure 3 shows the retainment ratios. We make the following observations:

**VS needs in-dialect data.** The strongest reduction is measured when leaving out the VS dialect. The drop is consistent across all three models. For XLS-R, the BLEU score only retains 87% of the performance. *Trafo* only retains 78% of the original BLEU score, and *Whisper* retains 83% of the *All-dialects* score. This drop was expected, as the VS dialect is the one that differs the most from all the other dialects (cp. Figure 2).

**ZH and CS do not require much in-dialect data.** Other dialects, such as ZH or CS, experience a smaller reduction. In fact, with XLS-R, both dialects almost completely retain their performance, and with *Trafo*, they retain 92% and 94% of the performance on average, respectively. This finding matches the **CS/ZH-central-hyp** hypothesis since ZH and CS benefit the most from other dialects.

**Mutual dependence.** We note that when one dialect is left out with XLS-R, the other dialects do not suffer a performance loss. In some cases, the performance even exceeds the *All-dialects* setting, with ratios above 1. This is due to XLS-R's extensive pre-training, which allows it to adapt to new languages with relatively small amounts of data. Thus, the results with *Trafo* are more revealing. When VS is left out, the other dialects suffer almost no deterioration, indicating the special status of this dialect, which is in line with the **VS-hyp** hypothesis. For the other dialects, omitting them leads to a greater performance loss on the other dialects. The largest overall drop in performance occurs when BS or ZH are left out, where the average retainment ratio of the other dialects is at 92%. For BS, this drop is not expected, as BS does not have as high pairwise similarities as ZH or CS. For

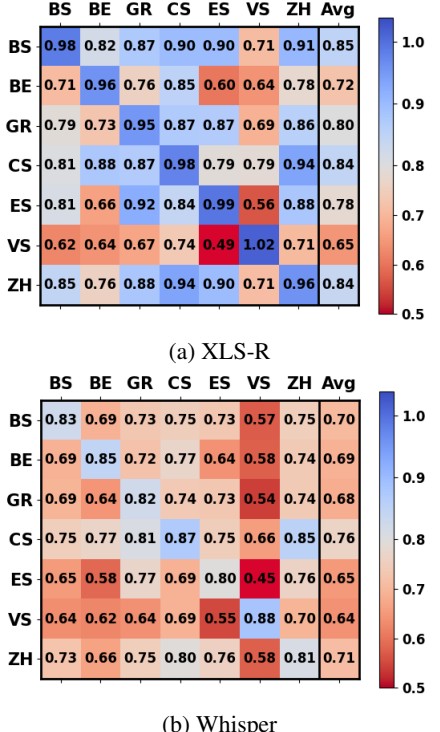

(a) XLS-R

(b) Whisper

Figure 4: Results of SD Experiment, when the system is trained on the dialect of the row. The last column shows the average score of the ratios (the average excludes the diagonal values.)

ZH, the decrease is consistent with the centrality hypothesis of ZH. However, there are no pairs of dialects where omitting them leads to a loss of more than 10% of the original performance.

## 5.2 Single Dialect Fine-Tuning

The single dialect fine-tuning (SD) experiment fine-tunes the models on a single dialect and measures the effect on all dialects. Table 2 gives an overview of the results. We can see that the final BLEU scores for *Trafo* were 0, as 50 hours of data are too little to train such a model from scratch. Thus, we only show and discuss the XLS-R and *Whisper* retainment ratios in Fig 4.

**Training on a single dialect is sufficient for that dialect.** The results show that with XLS-R, single dialect training leads to at least 95% of the *All-dialects* performance.

**VS Status confirmed.** The SD experiments again confirm VS's status, as the most distinct dialect as training on VS alone leads to an improvement of 2% over the *All-dialects* setting. We also note that VS and ES have the highest distance in both the XLS-R and the *Whisper* experiments, which corresponds to the linguistic distance in Figure 2.

**ZH is a central dialect.** The SD experiments also confirm that the similarity of ZH to ES and CS is reflected in the empirical results. When training on ZH only, the CS and ES scores retain at least 90% (for *Whisper* the retainment is also at least 76%). Vice versa, we observe the same pattern. The similarity between ZH and BS is less pronounced with both XLS-R and *Whisper*, which is in line with Figure 2. As with the LOO experiments, BS-only also reaches a high average score, which is not expected from the hypotheses.

**BE is dissimilar to both GR and ES.** The SD experiments also confirm the **BE/ES-GR-hyp** hypothesis. For systems trained only on BE, the GR and ES scores suffer the most (next to VS). Also, when the system is trained on either GR or ES, the BE scores suffer the most (next to VS).

Overall, the experiments confirm the hypotheses that we derived from the dialect distances based on the DDAGGS data. Most notably, the difference of VS to all the other dialects is as pronounced as the hypotheses predicted. Also, the status of ZH as a central dialect is confirmed by our experiments. On the model side, we note that both XLS-R and *Whisper* have a strong pre-training, where leaving out one dialect does not hurt performance. On the other hand, training only on a single dialect leads to subpar performance on samples of a different dialect.

## 6 Swiss German differences to Standard German

We investigate two main differences between Swiss German and Standard German (cp. Section 2.2). First, the usage of the past tense, i.e., samples where the Standard German text includes the preterite (simple past), which is not used in Swiss German. Second, the usage of words which are different in Swiss German compared to Standard German.

### 6.1 Preterite

We expect to see a mismatch between the transcript and the reference in those samples where the Standard German reference contains the simple past tense. We applied spaCy[7] to find samples in the test set where the preterite tense is used in the Standard German text. There are 5908 samples containing preterite tense (23.4%). We used the *All-dialects* model to compute the BLEU scores separately for

---

[7] https://spacy.io

| Text Type | Correct | Response |
|---|---|---|
| TARGET | - | der gemeinderat hatte hier bereits auf fr 10000000 gekürzt |
| REWRITTEN | - | der gemeinderat hat hier bereits auf fr 10000000 gekürzt |
| HYPOTHESIS | Yes | der gemeinderat hat hier bereits auf 100 000 franken gekürzt |
| TARGET | - | wir wurden sehr gut informiert und dokumentiert |
| REWRITTEN | - | wir sind sehr gut informiert und dokumentiert worden |
| HYPOTHESIS | No | wir sind sehr gut informiert und dokumentiert |
| TARGET | - | der 42 jährige hatte dort heimlich trainiert |
| REWRITTEN | - | der 42 jährige hatte dort heimlich trainiert |
| HYPOTHESIS | ? | der 42 jährige trainierte dort heimlich |

Table 3: Examples of Target, Rewritten Target, and generated Hypothesis for the Past Tense experiments.

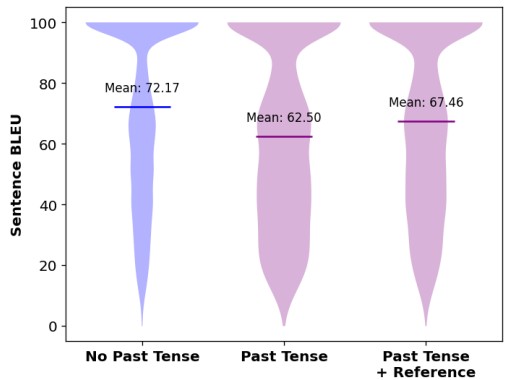

Figure 5: Comparison of distributions of sentence level BLEU scores for sentences with (Past Tense) and without past tense (No Past Tense). The Past Tense + Reference shows the BLEU distribution for samples with the rephrased reference. The differences between "No Past Tense" and "Past Tense" are significant according to Welch's t-test (p=5.95e-104).

the samples with past tense and those without past tense. Figure 5 shows the results. Samples with preterite tense in the Standard German text perform significantly worse than the ones without (mean BLEU of 0.625 vs. 0.722).

**Qualitative Analysis.** A qualitative analysis revealed two types of errors among the samples containing preterite tense in the target. The first category of mistakes is those that could not be traced back to the use of past tense, i.e., those where other mistakes yielded a lower BLEU score.

In the second category, the mistakes are due to the past tense. There, we noticed that the generated text often did not match the tense of the hypothesis. Thus, following the idea of paraphrasing to extend the set of references (Paonessa et al., 2023), we extended the set of references for all the past tense samples by translating them into the past perfect form. For this, we used ChatGPT[8]. In Figure 5, the

impact is shown (Past Tense + Reference). Extending the references yields an improvement of almost 5 points in BLEU score, i.e., there are 1197 out of 5908 samples where the BLEU score improves when adding the rewritten reference.

This result shows that measuring the correctness of samples with past tense is not trivial since there are two cases to consider: 1) the transcript is correct but uses the past perfect form, which does not match the reference. 2) the transcript is wrong due to difficulties handling the past tense. In Table 3, we show some examples. In the first example, the hypothesis uses the past perfect form correctly, while the target uses the past perfect. In the second example, which is in the passive voice, the hypothesis uses the present perfect but omits the participle. In the last example, the hypothesis uses the simple past while the target uses the past perfect. These examples illustrate the difficulty of measuring the effects of past tense.

### 6.2 Vocabulary

We collect lists of words where Swiss German and Standard German are likely to differ. We then measure the differences in BLEU scores between samples that contain such words and those that do not. To our knowledge, there is no inventory of Standard German words which are realised differently in Swiss German dialects. Therefore, we use three data sources as a proxy and apply (semi-)manual filtering to each.

1) **Volumes 5-8 of the DDAGGS** contain 64 lexical features that are expected to differ between the different dialects. We keep those entries where none of the realisations are cognates of the Standard German equivalent - e.g., we keep "heraus" ("out" in the directional sense) with realisations "us", "use", "usi", "uss", "usse", "uus", "uuse", "uusi" and "uuss", but we discard "flicken" ("to mend") with realisations "flicke", "nääje", "büeze",

---

[8]We used the gpt-3.5-turbo model, https://platform.openai.com/docs/guides/gpt

| Text Type | Correct | Response |
|-----------|---------|----------|
| TARGET | - | etwas weniger als eine stunde war er dort gewesen |
| HYPOTHESIS | Yes | etwas weniger als eine stunde war er da |
| TARGET | - | und wie schaut es aus beim direkten vergleich |
| HYPOTHESIS | Yes | und wie sieht es aus beim direkten vergleich |
| TARGET | - | ich arbeite sehr oft mit diesen werten |
| HYPOTHESIS | Yes | ich arbeite sehr viel mit den werten |

Table 4: Examples of Targets and generated Hypotheses for the vocabulary experiments.

etc. After this filtering step, 18 items remain.

2) **The Wikipedia entry on Helvetisms**[9] contains 503 entries representing lexical items specific to Swiss (Standard) German. We remove proverbs and obsolete expressions and then manually filter the remaining entries: we try to intuit whether a given Standard German word is usually realised in Swiss German with a different lexeme (because the Standard German lexeme does not exist or is rare/dispreferred). This is a subjective process that is limited by the evaluator's knowledge and perception of the different dialects. After filtering, 262 words remain.

3) **The GSWNorm 2022 Shared Task** (von Däniken et al., 2022) presented a data set of 7284 Swiss German sentences (106K tokens / 20K types) with word-level normalisation into Standard German. In order to reduce the set of types to be filtered manually, we apply a heuristic: we keep only those pairs where the first letters of the two words are not identical, leading to 2569 candidate types. We then filter these in the same way as Wikipedia Helvetisms, resulting in a final list of 267 items. We combine these three word lists and after eliminating duplicates between the lists, 522 vocabulary items remain. There are 2975 samples (12.1%) that contain at least one special vocabulary item.

**Results.** Figure 6 shows the difference in BLEU score between those samples without special vocabulary (No Special), the samples with special vocabulary (Special), and *All-dialects* (All). The difference in BLEU score is large (66.13 vs. 70.39). However, as with the Past Tense, the source of the BLEU difference is unclear, i.e., we do not know whether the difference stems from the transcript being wrong or the hypothesis using a synonym not covered by the reference (such synonyms may be introduced during translation and recording, see Section 4.1). To illustrate this issue, Table 4 shows examples of mismatches between the transcript and

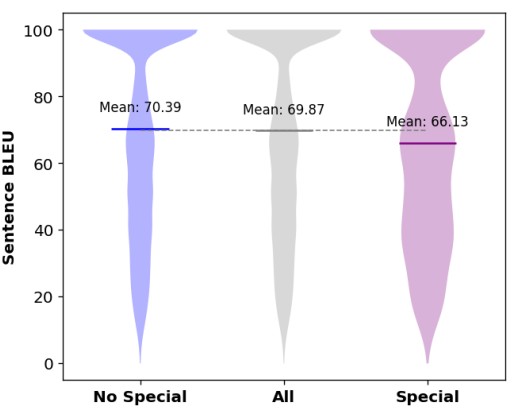

Figure 6: Comparison of distributions of sentence level BLEU scores for sentences containing words of special vocabulary. The differences between "No Special" and "Special" are significant according to Welch's t-test (p=5.3e-13).

the target. In most examples, the Swiss version of the word was used. For instance, Swiss German "wie *schaut* es aus", vs. Standard German "wie *sieht* es aus". Thus, a fair amount of the mismatch in terms of BLEU score can likely be attributed to using the Standard German version in the target.

## 7 Conclusion

Our findings show that the empirical results are consistent with the linguistic distances between the dialects and the differences between Swiss German and Standard German. For example, dialects similar to other dialects (such as ZH) positively affect others when included in the training data. In contrast, dialects that differ from others (such as VS) need in-dialect data to perform well. The usage of preterite and vocabulary specific to Swiss German impact the BLEU score negatively. Thus, future work is concerned with investigating methods dealing with more efficient dialect transfer and handling the differences between Swiss German and Standard German.

[9] https://de.wikipedia.org/wiki/Liste_von_Helvetismen#Wortschatz, accessed 12 May 2023

## Limitations

**No Learning Curve.** In our experiments, we either used the full set of dialect data (SD) or none of the dialect data (LOO). A more fine-grained analysis would include computing learning curves to understand how much "new" dialect data is needed. This would also give more hints on the interplay between dialects. However, these experiments would be very costly to run.

**No Tests on very large SOTA models.** We limited our experiments to models with around 300M parameters and not very large billion-parameter models. There are two reasons: first, the training time and cost would have become prohibitively large, and second, as shown in Appendix A, the insights are expected to be largely the same.

**Error Attribution Past Tense.** Measuring the mistakes caused by the past tense is not trivial since the mistakes could have two main sources: 1) a non-tense-related error or 2) a tense-related error. From the latter, there are again two subcases: 1) the model could not handle the tense and made mistakes due to that (e.g., second example in Table 3) 2) the model behaves well, but the reference does not match the tense generated by the model (which could be caused by the translation and recording process, see Section 4.1). Thus, we cannot measure how many errors are due to which of the above error types, only the impact on the BLEU score.

**Error Attribution Vocabulary.** Similarly, for the special vocabulary, we can only measure the impact on the BLEU score. The errors could also be due to a non-vocabulary-related source. If they are caused by vocabulary, it is still not clear whether the error stems from using a wrong word or a synonym that is not covered by the reference.

**Word List Subjectivity.** The creation of the word list was done mostly ad-hoc and based on the subjective interpretations of the word-list creator. Furthermore, we did not differentiate between dialects and some dialects may use vocabulary similar to Standard German.

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

## A Comparison XLS-R 1B model vs XLS-R 300M model

In Figure 7, we compare the XLS-R-1B and the XLS-R-300M models. We trained both models on

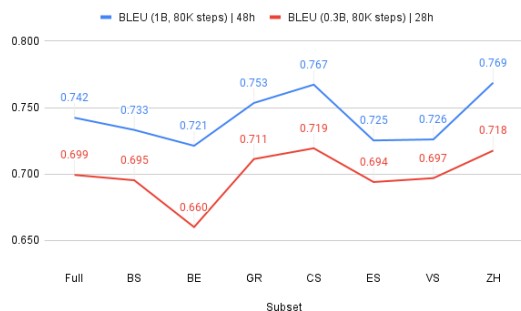

Figure 7: Comparison of two XLS-R models: XLS-R 1B and XLS-R 300M.

the STT4SG-350 balanced dataset for 80K steps. The 1B model took 48 hours to complete the training, while the 300M parameter model took 28 hours. The 1B model obtains a higher BLEU score on all dialects in the test set. Both models have the same strengths and weaknesses in the different dialects. Therefore, our findings on the 300M model can be transferred to the 1B parameter model. This study uses the 300M model as we aim for shorter training times.

## B Comparison of XLS-R 300M with CTC vs. sequence-to-sequence decoder

We executed all XLS-R-300M experiments with two different decoders: The standard CTC decoder and a sequence-to-sequence (Seq2Seq) decoder. In Table 5 we compare the two decoders on the STT4SG-350 balanced dataset. The CTC obtains an overall BLEU score of $0.709$, whereas the seq2seq decoder achieves $0.659$. In the low data regime with the dialect from scratch experiments, the CTC model results in an average $0.576 \pm 0.044$ BLEU score and seq2seq in an average $0.107 \pm 0.026$ BLEU score. The average BLEU score is calculated over seven training runs using a different dialect for training each time. We observe, the decoder in the Seq2Seq architecture cannot learn decoding with only 34 hours of training data.

## C Linguistic Distance Matrices by Feature Type

In Figures 8a, 8b and 8c we show the linguistic feature matrices by feature type. While Figure 2 shows the Relative Identity Values (RIV) across all 289 features of the DDAGGS, they are presented individually here: Figure 8a shows the values based on the 107 phonological features, Figure 8b based

| Model | Full | BS | BE | GR | CS | ES | VS | ZH |
|---|---|---|---|---|---|---|---|---|
| CTC | 0.709 | 0.698 | 0.684 | 0.727 | 0.729 | 0.695 | 0.706 | 0.725 |
| Seq2Seq | 0.659 | 0.645 | 0.635 | 0.675 | 0.676 | 0.651 | 0.663 | 0.671 |

Table 5: Comparison of two XLS-R models: XLS-R with CTC decoding and XLS-R with Seq2Seq head.

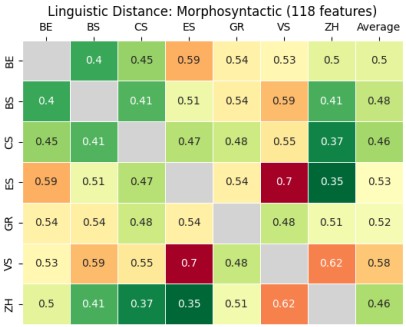

(a) Matrix of linguistic distance between dialect regions: phonological features

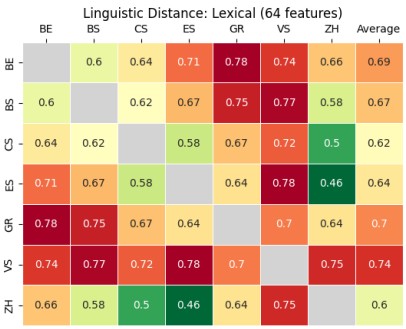

(b) Matrix of linguistic distance between dialect regions: morphosyntactic features

(c) Matrix of linguistic distance between dialect regions: lexical features.

Figure 8

on the 118 morphosyntactic features and Figure 8c based on 64 lexical features. We can see that VS is the most distinct dialect across all feature types, followed by GR, BE and ES in changing order changing depending on the feature.

## D   Experiment Details

The vocabulary used to preprocess the sentences is limited to lower-case characters and the German umlauts ä, ö, and ü. All characters with other accents are transformed into their corresponding character without accents, and hyphens are replaced with a space.

**XLS-R.** We use the fairseq implementation[10] and replicate the training procedure and model settings from (Plüss et al., 2023a). The runs on the full dataset and the Leave-One-Out (LOO) experiment are trained for 80K steps. The Single Dialect Fine-Tuning (SD) models are only trained for another 20K steps without any freezing during the warmup phase. All the final models correspond to the checkpoint with the best Word Error Rate on the validation dataset during training. There is no task-specific fine-tuning of the hyperparameters. The training for the 300M version of the model is conducted on 2 NVIDIA A100 40 GB GPUs. The 1B *All-dialects* model is trained on 4 NVIDIA A100 40 GB GPUs.

**Trafo** We replicate the model architecture and training procedure from Plüss et al. (2022). This model is based on the FAIRSEQ S2T library (Ott et al., 2019; Wang et al., 2020). The runs on the full dataset and the Leave-One-Out (LOO) experiment are trained for 80K steps. The Single Dialect Fine-Tuning (SD) models are only trained for another 10K steps. There is no task-specific fine-tuning of the hyperparameters. The models are trained on a single NVIDIA A100 40 GB GPUs.

**Whisper** We use the Huggingface (Wolf et al., 2019) implementation[11] of the Whisper small model including the provided fine-tuning procedure. The runs on the full dataset and the Leave-One-Out (LOO) experiments are trained for 80K steps and a warmup phase of 10K steps. The Single Dialect Fine-Tuning (SD) models are only trained for another 10K steps with a warmup phase of 1K steps. The learning rate is set to $1e - 5$. There is no task-specific specific fine-tuning of the hyper-

---

[10] https://github.com/facebookresearch/fairseq/tree/main/examples/wav2vec/xlsr
[11] https://huggingface.co/docs/transformers/model_doc/whisper

| Model | Metric | Full | BS | BE | GR | CS | ES | VS | ZH |
|---|---|---|---|---|---|---|---|---|---|
| XLSR | WER | 16.1 | 16.6 | 18.3 | 15.0 | 14.7 | 16.9 | 16.2 | 14.9 |
| - | chrF | 88.6 | 88.1 | 87.0 | 89.5 | 89.8 | 88.0 | 88.5 | 89.5 |
| - | CER | 8.3 | 8.2 | 9.9 | 7.8 | 7.3 | 9.0 | 8.2 | 7.6 |
| - | TER | 15.8 | 16.4 | 18.0 | 14.7 | 14.5 | 16.6 | 16.0 | 14.6 |
| Trafo | WER | 24.1 | 26.1 | 29.4 | 21.8 | 20.5 | 23.5 | 25.1 | 22.5 |
| - | chrF | 82.7 | 81.2 | 79.0 | 84.6 | 85.5 | 83.3 | 81.7 | 84.0 |
| - | CER | 14.0 | 14.7 | 17.8 | 12.2 | 11.6 | 13.7 | 14.6 | 13.0 |
| - | TER | 23.9 | 26.0 | 29.2 | 21.5 | 20.3 | 23.2 | 25.0 | 22.3 |
| Whisper | WER | 25.0 | 26.5 | 27.5 | 23.6 | 23.3 | 25.4 | 25.2 | 23.6 |
| - | chrF | 86.5 | 85.4 | 84.6 | 87.4 | 88.1 | 86.4 | 86.3 | 87.6 |
| - | CER | 23.7 | 24.4 | 24.8 | 23.3 | 22.3 | 23.6 | 24.4 | 22.9 |
| - | TER | 22.8 | 24.2 | 25.2 | 21.5 | 21.1 | 22.9 | 23.0 | 21.4 |

Table 6: BLEU scores of all experiments. Each experiment is a tuple of model and experiment type. There are three types: All-dialects, Leave-One-Out, and Single Dialect. For each setting, we report the score achieved on each dialect separately, and the average score achieved on the full test set.

parameters. The models are trained on 4 NVIDIA A100 40 GB GPUs.

# E    Experiments - Alternative Metrics

In the main text, we use the BLEU score for our analysis, here, we present the same type of results using different metrics: WER, TER and chfF. Figure 9 shows the LOO experiment results using WER, figure 10 shows the LOO experiment results using TER, and figure 11 shows the LOO experiment using chrF. All the metrics yield the same underlying results and conclusions as using the BLEU score. In table 6, the All dialects scores for the different metrics are presented. Figure 12 shows the SD experiment results using WER, figure 13 shows the SD experiment results using TER, and figure 14 shows the SD experiment using chrF. All the metrics yield the same underlying results and conclusions as using the BLEU score.

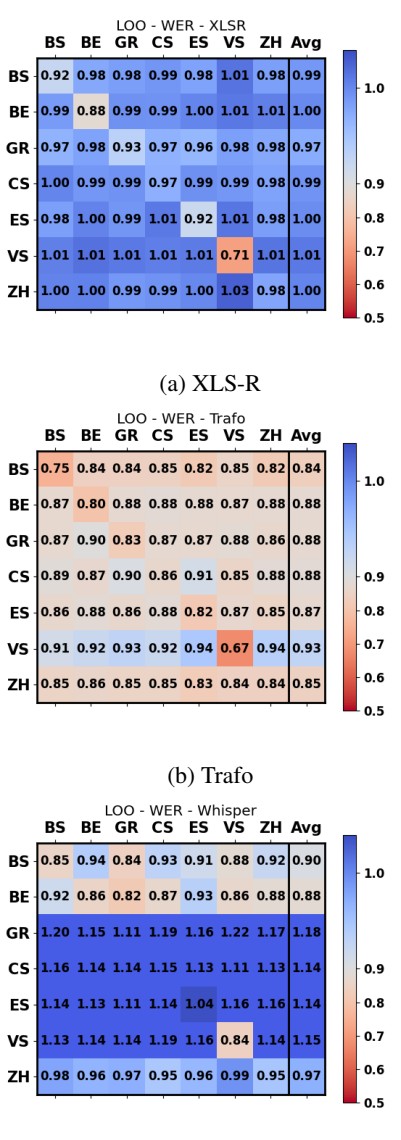

(a) XLS-R

(b) Trafo

(c) Whisper

Figure 9: Results of LOO Experiment: heatmap with retainment ratio. Each row shows the scores achieved for each dialect when leaving out the dialect of the row. The last column shows the average score of the ratios (the average excludes the diagonal values.)

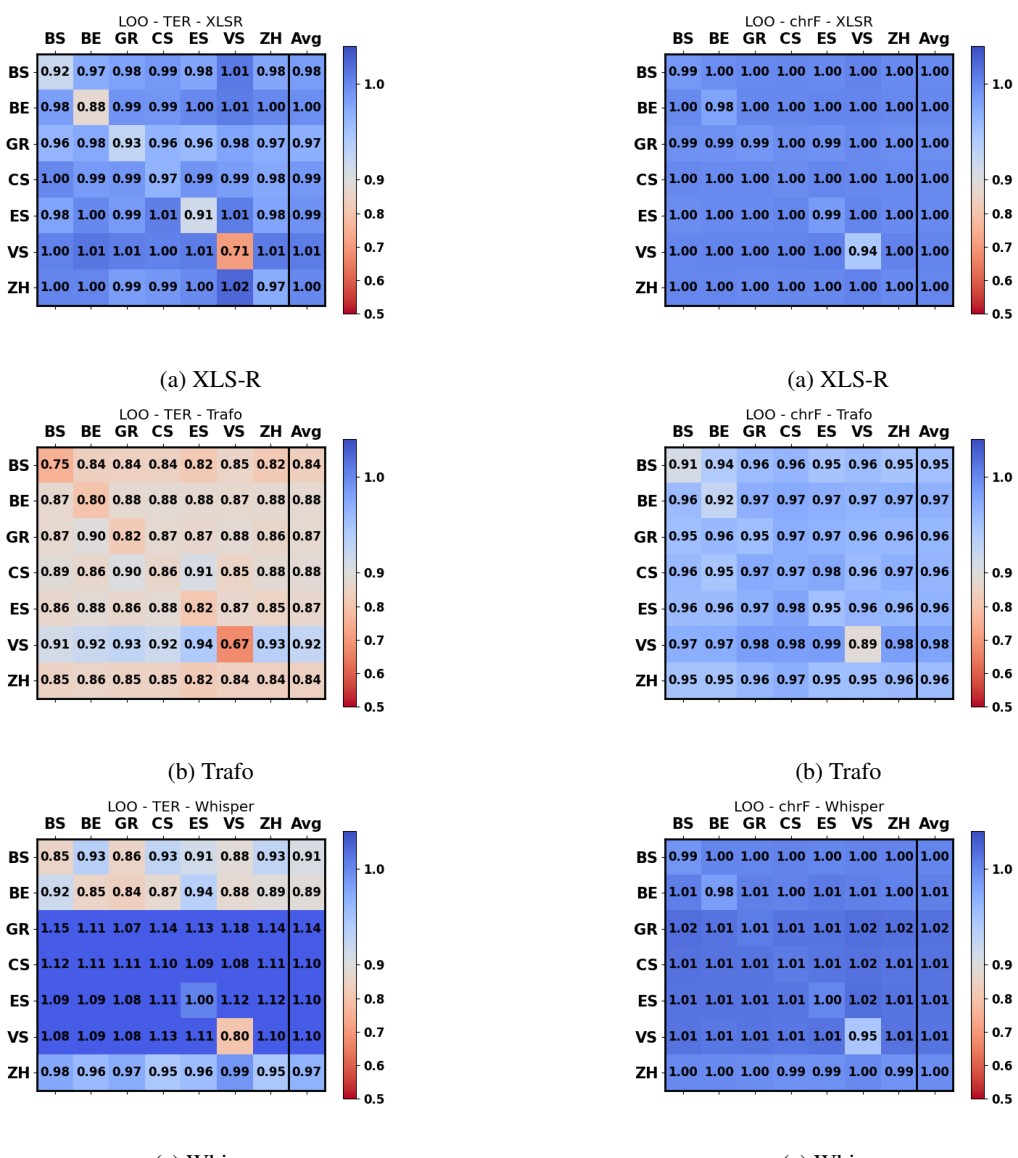

(a) XLS-R

(b) Trafo

(c) Whisper

Figure 10: Results of LOO Experiment: heatmap with retainment ratio. Each row shows the scores achieved for each dialect when leaving out the dialect of the row. The last column shows the average score of the ratios (the average excludes the diagonal values.)

(a) XLS-R

(b) Trafo

(c) Whisper

Figure 11: Results of LOO Experiment: heatmap with retainment ratio. Each row shows the scores achieved for each dialect when leaving out the dialect of the row. The last column shows the average score of the ratios (the average excludes the diagonal values.)

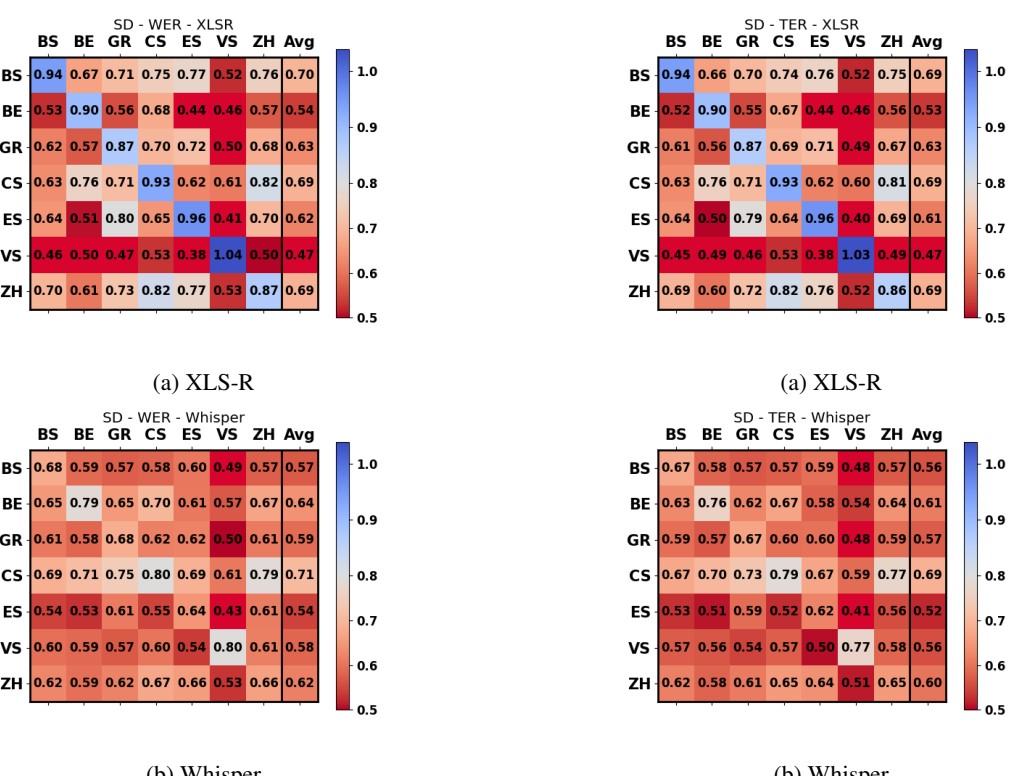

(a) XLS-R

(a) XLS-R

(b) Whisper

(b) Whisper

Figure 12: Results of SD Experiment, when the system is trained on the dialect of the row. The last column shows the average score of the ratios (the average excludes the diagonal values.)

Figure 13: Results of SD Experiment, when the system is trained on the dialect of the row. The last column shows the average score of the ratios (the average excludes the diagonal values.)

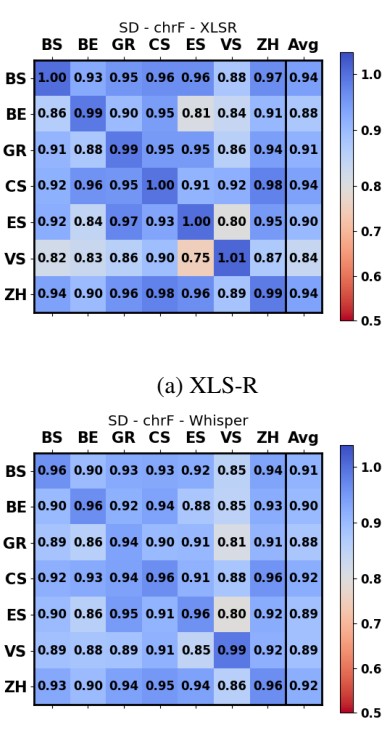

(a) XLS-R

SD - chrF - Whisper

|  | BS | BE | GR | CS | ES | VS | ZH | Avg |
|----|----|----|----|----|----|----|----|-----|
| BS | 0.96 | 0.90 | 0.93 | 0.93 | 0.92 | 0.85 | 0.94 | 0.91 |
| BE | 0.90 | 0.96 | 0.92 | 0.94 | 0.88 | 0.85 | 0.93 | 0.90 |
| GR | 0.89 | 0.86 | 0.94 | 0.90 | 0.91 | 0.81 | 0.91 | 0.88 |
| CS | 0.92 | 0.93 | 0.94 | 0.96 | 0.91 | 0.88 | 0.96 | 0.92 |
| ES | 0.90 | 0.86 | 0.95 | 0.91 | 0.96 | 0.80 | 0.92 | 0.89 |
| VS | 0.89 | 0.88 | 0.89 | 0.91 | 0.85 | 0.99 | 0.92 | 0.89 |
| ZH | 0.93 | 0.90 | 0.94 | 0.95 | 0.94 | 0.86 | 0.96 | 0.92 |

(b) Whisper

Figure 14: Results of SD Experiment, when the system is trained on the dialect of the row. The last column shows the average score of the ratios (the average excludes the diagonal values.)