# OpenReview forum: "Dialect Transfer for Swiss German Speech Translation"
_EMNLP/2023/Conference — EMNLP 2023 Findings_

### Official Review · Reviewer_LM7U · 2023-08-02

**Soundness:** 4

**Excitement:**

4: Strong: This paper deepens the understanding of some phenomenon or lowers the barriers to an existing research direction.

**Paper Topic And Main Contributions:**

The paper focuses on translating from Swiss German Speech to text in high German. There are many thorough experiments regarding the influence of different dialects of Swiss German on each other, and how some dialects are more distinct and thus are more helpful when data from a dialect like that is used in training.  Three model architectures are tried out, and a very thorough analysis of the experimental results is provided.



**Questions For The Authors:**

The BLEU scores that you report are in general rather high, more than 50% on average. This means that there is quite a large overlap between the translation and the reference, which means that the task could be considered more an ASR task vs. MT task. Could you provide also WER (word error rate) and TER (translation edit rate) scores at least for some experiments? These would give insights: how different is translation from speech in Swiss German into High German text in terms of WER from the case where the speech and text are in the same language? Also, how much re-ordering is going on (difference between WER and TER numbers will tell us).

Also, if you compute WER and TER, you can also obtain edit distance alignment between words from the hypothesis and the reference, and this would help with the attribution of vocabulary errors (for example, you can directly look to which words the preteritum words are aligned, etc.)

Another question: assuming that you would have had recordings of the same utterances (from the test set) but spoken in High German directly, how high do you think would WER be? Would it be close to 0 given the recording conditions? An experiment like that could be done by using a TTS system to pronounce the utterances. This would give a kind of a lower bound on the quality of "translation" from Swiss German (I put translation in quotes here to highlight that at least for some of the spoken words it is speech recognition, not translation if they have a one-to-one correspondence with their written high German counterparts).

Related to this, if you fine-tune a pre-trained model like XLS-R or Whisper on Swiss German, would its performance degrade on High German? This is an important practical question, since, as far as I know, there is a lot of code switching going on in Switzerland between the two, and adding High German to the mix of Swiss German dialect is a good idea for practical applications of your MT/ASR system.




**Reasons To Accept:**

The experiments are very thorough, well-described, and well-justified. The paper will be of value for researchers who not only work on Swiss German but also on other languages and dialects without a standard written form.

The authors show an example of good scientific work, with a deep analysis of the experiments, focusing on specific phenomena like the past tense in Swiss German vs. high German, showing examples and trying to find explanations for the automatic scoring results.

Three models of different complexity are fine-tuned or trained from scratch, one of them being Whisper, which is the latest pre-trained speech model in fashion at the moment. However, it is interesting to see that another, smaller model XLS-R lead to better results. Also, the authors showed that it is sufficient to use a version of this model with fewer parameters to arrive at similar conclusions, thus performing experiments with less computational resources.

The paper is overall very clearly written and is easy to follow.



**Reasons To Reject:**

I don't see any major reasons for a rejection apart from maybe more ground-breaking work by other researchers that is more novel. In this paper, the novelty arises from the availability of a new dataset plus the excellent experimental methodology itself.

I would like the authors to address my questions below, though, especially regarding the computation of WER/TER which puts their experiments in the context of speech recognition as opposed to speech translation (and in my opinion, it is necessary to do so given the significant lexical and word order similarity of Swiss German and High German, despite their differences).


**Reproducibility:**

3: Could reproduce the results with some difficulty. The settings of parameters are underspecified or subjectively determined; the training/evaluation data are not widely available.

**Reviewer Confidence:**

4: Quite sure. I tried to check the important points carefully. It's unlikely, though conceivable, that I missed something that should affect my ratings.

**Typos Grammar Style And Presentation Improvements:**

Line 373: "to fine-tune such a model from scratch" - this phrase is contradictory - is it fine-tuning or training from scratch?
Table 3: this Table is not referenced in the main text of the paper, at least I could not find any reference to it.
Caption of Figure 5: are you sure about the p= ... of the statistical significance here, is it really this low, or is it a typo?

---

> ### Author Rebuttal · Authors · 2023-08-25
>
> Thank you very much for your comments and interesting questions. We will address them here.
>
> On using WER and TER
> * It is true that the BLEU scores are rather high, and might not be the ideal measure. However, using the WER is too strict since we are dealing with two language varieties and leads to having many false negatives. Generally, it is a hard setting to evaluate. Furthermore, we are interested in the relative differences in scores for the different scenarios. Having said that, we will gladly add WER and TER scores to the appendix. We will create the same types of Figures and will add the analog to Table 2 into the Appendix.
> * We computed the WER and TER for all our experiments. Regarding the question of the difference between WER and TER: it is small, with TER being only slightly lower.
> * * XLS-R Baseline: WER 16.07 vs. TER 15.82
> * * Trafo Baseline: WER 24.14 vs. TER 23.92
> * In order to keep the response relatively short, we show the WER scores for the pair VS and ZH (the rest will be added to the Appendix):
> *  * WER for the XLS-R Baseline: 16.07 overall, for VS 16.18 for ZH: 14.92
> * * WER for XLS-R leaving out VS on VS: 22.67 for ZH: 14.72 -> thus, confirming that VS needs in-dialect data and ZH even benefits from lack of VS data.
> * * WER for XLS-R with VS from scratch for VS: 15.62, and ZH: 29.75 again confirming the VS statement
> * We will add the plots for WER, TER, and chrF in the Appendix. However, the main conclusions remain the same under these other metrics as well.
>
> On the performance of Swiss German models on Standard German.
> * We ran both types of experiments that you asked for.
> *  * Performance of Standard German on XLS-R trained on Swiss German.
> * * Change in performance of Whisper after Swiss German fine-tuning.
> * To evaluate the performance of the Swiss German model on Standard German, we leverage the CommonVoice 11 DE data (the TTS route is a bit out of scope in this short timespan). Note that we used the XLS-R 1B model and additional data for Swiss German (SG4STT + SDS-200 + SPC), we report those results (we ran them already some weeks ago). This is why the BLEU scores for Swiss German are much higher than in the paper. We show 4 scenarios 2 using XLS-R and 2 using Whisper:
> * * XLS-R
> * * * (1) Fine-tune XLS-R on Swiss German data. Results: Common Voice 11 DE testset BLEU of 46.31 (WER: 36.30) & STT4SG-350 test: 76.69 (WER: 12.81)
> * * * (2) Fine-tune XLS-R on Swiss German data+CommonVoice Train.  Results: Common Voice 11 DE testset BLEU of  85.09 (WER: 8.04) & STT4SG-350 test: 75.28 (WER: 13.53)
> * * Whisper
> * * * (1) Whisper Performance on CommonVoice Test before fine-tuning on STT4SG. Results: Common Voice 11 DE testset BLEU of 64.55 (WER: 22.48) & STT4SG-350 test: 26.03 (WER: 51.27)
> * * * (2) Whisper Performance on CommonVoice Test after fine-tuning on STT4SG. Results:  Common Voice 11 DE testset BLEU of 18.93 (WER: 69.42) & STT4SG-350 test: 62.93 (WER: 25.01)
> * Thus, we see the following insights:
> * * Training only on Swiss German does not suffice to learn a good Standard German model.
> * *  Training on a mix of Standard German and Swiss German yields good results for both languages with a slight decrease in Swiss German performance.
> * *  Fine-tuning Whisper on Swiss German lets it forget all about Standard German (catastrophic forgetting). On the other hand, the pre-trained Whisper performs reasonably well on Standard German, but very poorly on Swiss German.
> * *  The results indicate that there is quite a difference between Standard German and Swiss German.
> * *  We’ll add this to the Appendix
>
> On the difference between ASR vs. ST/MT.
> * The issues that you raise are very interesting. That is, is our task to be interpreted better as a Speech Translation or ASR task? Recently, it was mostly framed as Speech Translation since Swiss German words require a translation and some reordering as well. However, as you rightly point out, it is not as different as French is to English. We think that this would be a very interesting topic for a separate followup paper.
> * We think that for the purpose of our work, the setting of Speech Translation and using BLEU scores suffices, since also using WER or TER yield the same overall conclusions and outcomes as the BLEU analysis - however, we will report them in the Appendix.
>
> Various things that we will update.
> * On the p-value in Figure 5. Yes, this is the statistical significance, which in our case is significantly below the p < 0.05 value.
> * You are right, we will reference Figure 3 in Section 6.2
> * When talking about “from scratch” we will normalize it to “train from scratch”.

---

### Official Review · Reviewer_xjs8 · 2023-08-05

**Soundness:** 3

**Excitement:**

3: Ambivalent: It has merits (e.g., it reports state-of-the-art results, the idea is nice), but there are key weaknesses (e.g., it describes incremental work), and it can significantly benefit from another round of revision. However, I won't object to accepting it if my co-reviewers champion it.

**Paper Topic And Main Contributions:**

This paper addresses transfer between Swiss German dialects in speech translation from Swiss German to Standard German.
Specifically, it looks at translation of 7 different Swiss German dialects to German using the STT4SG-350 corpus with the balanced train splits of 34 hours per dialect. Compared are directly finetuning pretrained models (XLS-R and Whisper) on all dialects on a single dialect, or leave one out; and finetuning a transformer model possibly pretrained on SDS-200 (a Swiss German speech translation corpus) on each of these conditions (called "Trafo"). Two different model sizes are used for XLS-R to check whether model size has an effect (conclusions are shown to be robust across model scales).
Results suggest that transfer ability roughly corresponds to linguistic distances between dialects derived from DDAGGS features.
Analysis on the preterite and vocabulary likely to be specific to Swiss German using crawled word lists with semi-manual filtering is conducted (each leading to the conclusion that there is a difference in these two cases, but stated that it is hard to pinpoint why).
Results suggest that leave one out is typically similar to including all dialects, but with a weaker pretrained model (Trafo) more distantly related dialects suffer as more data is left out, and finetuning on a single dialect is insufficient in all cases. There is a greater gap between all/leave one out/single dialect finetuning for Whisper than XLS-R (presumably because it is trained on less non-English data, and so more sensitive to having less finetuning data?).  Reasons for differences between XLS-R and Whisper are not discussed, and the comparison with Trafo is a bit murky.

**Questions For The Authors:**

Question A: Could you clarify what data the Trafo model was trained on? (pretraining and finetuning)

Question B: Given the other experiments showing CTC performs significantly better than a seq2seq decoder trained from scratch, why not use CTC here as well for a more appropriate comparison to a model trained directly on/for Swiss German? What is the size of the seq2seq decoder? Given that the target language is a high resource language (Standard German) why not use a pretrained decoder or additional language modelling data?

Question C: For context for the scores presented here, if a faithful dialectal *transcript* was output rather than the Standard German translation, what would the BLEU scores with the Standard German reference translation be? Given that the vocabulary analysis says some models leave special vocabulary in Swiss German, this would be particularly good to report.

Question D: Would it be possible to include Standard German in the linguistic distances for context?

Question E: What were the model vocabularies used? It is said that the preprocessing limits characters to lowercase characters (ASCII characters?) and the German umlauts ä, ö, and ü. Does this mean all (non-Whisper) models use character-level vocabularies, or is this just a normalization step?

**Reasons To Accept:**

- Experimental comparison of speech translation of Swiss German dialects into Standard German, with three different models, and three different transfer/finetuning settings (all, some, one)
- Detailed analysis of differences between dialects with examples

**Reasons To Reject:**

- Narrow scope using existing models and techniques; differences between dialects are the focus, rather than the techniques, making the conclusions potentially helpful to a specific target audience but not necessarily generalizable to other cases, particularly given that there is no task-specific parameter tuning and not a direct comparison between model settings
- Some experimental details are missing or confusingly stated. A number of basic model/data details are referred to other works and should be included here (at least in the appendix) for clarity instead of requiring lookups in multiple other papers (for example, the size of the Trafo data, the model architecture, the size of the seq2seq decoder).
- The space could be used more effectively; analysis is at times redundant with the text as it lists what is in figures and tables (for example, significant space is used for heatmaps whose numbers which are then summarized in the text, rather than further discussion of why there may be differences in performance or trends such as between Whisper and XLS-R).

**Reproducibility:**

3: Could reproduce the results with some difficulty. The settings of parameters are underspecified or subjectively determined; the training/evaluation data are not widely available.

**Reviewer Confidence:**

4: Quite sure. I tried to check the important points carefully. It's unlikely, though conceivable, that I missed something that should affect my ratings.

**Typos Grammar Style And Presentation Improvements:**

It is not completely clear what data the Trafo model is trained on:
- Section 4.2 says 'the baseline of Plüss et al 2022 is replicated and fine-tuned on our data', and the balanced training is said to be 34 hours per dialect, and looking up Plüss et al 2022 it is a 200 hour Swiss German to Standard German corpus and the baseline is a 12-6 layer encoder-decoder trained on all of the data. On L373 it says re Trafo "50 hours of data are little to fine-tune such a model from scratch" when mentioning that the single dialect fine-tuning resulting in a score of 0.
- Is it the case that the Trafo model is pretrained on the 200 hours from SDS-200, and then finetuned on each condition? (all dialects, LOO, single dialect). Or, has it been trained from scratch on each condition, using the baseline architecture only from Plüss et al 2022, in which case these models have been trained on between 34 and 276 hours? Should L373 say "34 hours is too little to fine-tune such a model" or "34 hours is too little to train such a model from scratch" ? If they are in fact fine-tuned, it is surprising that with 200+ hours of Swiss German to Standard German data there are scores of 0?

Table 2: Consider renaming "Baseline" to "All dialects" for clarity and to help explain why in most cases it performs the best

BLEU scores are typically reported multiplied by 100 to be 0-100, and here presented 0-1; the presented scores are presumably quite high (45-73) and not very low (<1), consider multiplying by 100 to standardize.

It is stated that model checkpoints are chosen based on WER, but WER is not reported. Consider adding WER for context in the results table, or at least in the appendix.

BLEU is known to be less appropriate than other metrics such as chrF for dialects which are often marked by differences of 1 to 2 characters, causing potentially large differences in BLEU for small differences in system outputs; consider reporting at least chrF as an additional metric.

Why use the BLEU implementation from Plüss 2023b (which uses NLTK and is not common practice because it is not comparable to other implementations), and not SacreBLEU [(Post 2018)](https://aclanthology.org/W18-6319/)? It should be noted not just that BLEU is calculated using the script provided by (Plüss et al, 2023b) but include "which uses the NLTK implementation" so this is clear and scores are not compared with a different implementation.

Typo: L772 "task-specific specific"

---

> ### Author Rebuttal · Authors · 2023-08-25
>
> Thank you for your insightful questions, we will answer them here and address some of your concerns.
>
> Question A:
> * The Trafo model is a randomly initialized Encoder-Decoder model, which is trained from scratch. Meaning that there is no pre-training using SDS-200 or any other pre-training, only trained on the various settings in the paper (All-dialects, LOO, SingleDialect from the SG4STT corpus).
>
> Question B:
> * The experiments in Appendix B only focus on the XLS-R 300M model. In that case, using CTC performed better than adding a neural decoder.
> * Why not use a pre-trained decoder? In this work, we were more interested in the interplay between the dialects, and not in maximizing the scores. The Trafo case covers the setting where the model is trained completely from scratch without any pretraining. We only employed FairseqLM during the decoding as an additional LM. The Whisper experiments cover the setting of having a pre-trained encoder and decoder architecture.
> * Trafo details: 2-layer convolutional subsampler, 12 transformer encoder layers, 6 transformer decoder layers. Each transformer layer has 8 attention heads, and an embedding size of 512. This model corresponds to the T-Md (Transformer Medium) model from fairseq S2T (Wang et al., 2020) with 72 million parameters.
>
> Question C:
> * To check how well a Swiss German transcript would perform when compared to a Standard German Reference, we ran a new experiment where we used the SwissDial corpus (Dogan et al 2021) that contains parallel Data of different dialects and Standard German texts. We computed the BLEU score between the Dialect text and the Standard German texts for each dialect separately. The BLEU scores are very low and range between 1.3 (CS) and 8.6 (Grisons). The reason is that written Swiss German differs significantly from Standard German as 1) it is written roughly phonetically, 2) there are lexical differences (e.g. “stoßen…aus” vs “macheds” in the example), and 3) differences in word ordering. For example:
> * * Standard German: “Wenn sie sich versammeln oder berühren stoßen sie ein nasales Krächzen aus.”
> * * Zurich Dialect: “Wenn si sich versammled oder berüered macheds es nasals Chrächze.”
>
> Dogan-Schönberger, P., Mäder, J., & Hofmann, T. (2021). Swissdial: Parallel multidialectal corpus of spoken swiss german. arXiv preprint arXiv:2103.11401.
>
> Question D:
> * The (D)DAGS dataset that we use contains only Swiss German dialects but not Standard German, since the data was elicited in Switzerland.
> * In the literature, we found that Scherrer et al. (2019) calculate a dialectality measure for Swiss German on data from the ArchiMob dataset, which contains Swiss German recordings with dialectal and normalised transcriptions. In a simplified procedure, they calculate the Levenshtein distance between the dialectal transcripts and the normalisation layer of ArchiMob to assess the dialectality of the different respondents  (see Section 5.2.2 / Figure 9). The emerging picture is not very clear and does not correlate with geography.
>
> Scherrer, Yves, Tanja Samardžić, and Elvira Glaser. "Digitising Swiss German: how to process and study a polycentric spoken language." Language Resources and Evaluation 53.4 (2019): 735-769.
>
>
> Question E:
> * For XLS-R CTC, we used the lower-cased alphabet, the umlaute (ä,ö,ü), and the digits from 0-9 (i.e., string.ascii_lowercase + string.digits + 'äöü ')
> * For Whisper, we used the vocabulary provided by them.
> * For Trafo: 5000 Sentence Piece Model fitted on sentences normalized with the same vocab (only lowercase). To fit the sentence piece model, we used over 67 million sentences from Europarl v7 de, News crawl (WMT) 2019 de, Parlspeech v2 (Bundestag, Nationalrat)
>
>
> Other comments:
> * We will update the descriptions of the experiments in our paper to make the experimental setting clearer, especially the Trafo case.
> * We will update Baseline -> All Dialects, Report BLEU*100, and also put WER info into the Appendix (also requested by Reviewer LM7U).
> * Regarding BLEU by Pluss.  We used the Pluss version, since most papers on Swiss-German to Standard German use this script. We can clarify the use of the NLTK implementation. Furthermore, we were more interested in the relative values than the absolute ones. We also computed SacreBLEU, which yielded the same relative outcome, just with slightly higher scores.
> * Use of space: we believe that a discussion of the heatmaps is helpful to some readers since they compress a lot of information and scores into one number.  However, we will check whether this can be shortened.
> * Adding chrF: we computed the chrF scores for all experiments, and will add them into the appendix, alongside WER, and TER as requested by R-LM7U. In all cases, the effects that we measured using BLEU scores remain the same just with different magnitudes.

---

### Official Review · Reviewer_qYnP · 2023-08-05

**Soundness:** 5

**Excitement:**

4: Strong: This paper deepens the understanding of some phenomenon or lowers the barriers to an existing research direction.

**Paper Topic And Main Contributions:**

This paper sheds light on the Swiss German varieties and investigates the interplay of dialects vs. Standard German in speech translation. The methodology is compelling and the findings are very interesting. I particularly like the detailed description of various linguistic features of the targetted dialects.

**Reasons To Accept:**

I'd like this paper to be accepted because it goes beyond the current "macro-language" perspective in NLP and delves into dialects and non-standard varieties. Also, the low-resourced setting and how to deal with it is a major contribution to the field.

**Reasons To Reject:**

No reason.

**Reproducibility:**

4: Could mostly reproduce the results, but there may be some variation because of sample variance or minor variations in their interpretation of the protocol or method.

**Reviewer Confidence:**

4: Quite sure. I tried to check the important points carefully. It's unlikely, though conceivable, that I missed something that should affect my ratings.

---

> ### Author Rebuttal · Authors · 2023-08-25
>
> Thank you very much for the positive feedback, we appreciate this a lot.

---

### Meta-Review · Area_Chair_w98n · 2023-09-19

**Recommendation:** 4

**Metareview:**

This work addresses the problem of speech translation from spoken Swiss German dialects into written Standard German and investigates the performance in many different scenarios to demonstrate the effects of dialect diversity empirically.
The reviewers agree on its substance but do not fully acknowledge its “technical” contributions.

Here is a list of the pros and cons of this paper.
* Pros
- Thorough experimental comparison in different scenarios (xjs8, LM7U)
- Valuable for future studies in speech translation for low-resource languages/dialects without a standard written form (qYnP, LM7U)
- Detailed analyses and discussions on the differences among dialects (xjs8, LM7U)
* Cons
- Limited focus on Swiss German, which is not necessarily generalizable to other languages/dialects (xjs8); It is not so serious unless there are any studies working on very similar problems.

---

### Decision · Program_Chairs · 2023-10-07

**Decision:**

Accept-Findings

**Comment:**

This work addresses the problem of speech translation from spoken Swiss German dialects into written Standard German and investigates the performance in many different scenarios to demonstrate the effects of dialect diversity empirically.
The reviewers agree on its substance but do not fully acknowledge its “technical” contributions.

Here is a list of the pros and cons of this paper.
* Pros
- Thorough experimental comparison in different scenarios (xjs8, LM7U)
- Valuable for future studies in speech translation for low-resource languages/dialects without a standard written form (qYnP, LM7U)
- Detailed analyses and discussions on the differences among dialects (xjs8, LM7U)
* Cons
- Limited focus on Swiss German, which is not necessarily generalizable to other languages/dialects (xjs8); It is not so serious unless there are any studies working on very similar problems.